# Behavioral and social predictors of suicidal ideation and attempts among adolescents and young adults

**Yeganeh Shahsavar, Avishek Choudhury** [ORCID] *

Industrial and Management Systems Engineering, Benjamin M. Statler College of Engineering and Mineral Resources, West Virginia University, Morgantown, West Virginia, United States of America

* avishek.choudhury@mail.wvu.edu

## Abstract

Suicide is now the 2nd leading cause of death. Despite existing prevention initiatives, the incidence of suicide deaths among younger population has risen in recent years. The study aimed to determine the social and behavioral predictors of suicidal ideation (SI) and suicide attempts (SA) among adolescents and young adults. We analyzed data from the 2021 Youth Risk Behavior Surveillance System (YRBSS) involving 17,232 U.S. students in grades 9–12. Among respondents, 20.85% (3,593) reported suicidal ideation, and 10.17% (1,753) attempted suicide. Of the 3,526 participants with depression, 46.23% (1,630) experienced suicidal ideation, and 19.46% (686) attempted suicide. Among the 8,152 female participants, 27.42% (2,335) reported suicidal ideation, and 13.37% (1,090) reported suicide attempts. Also, 29.06% of 3,981 participants who were overweight experienced SI and 14.02% of them attempted suicide. We developed independent SI and SA prediction models using extreme gradient boosting (XGB), logistic regression (LR), and K-Nearest Neighbors (KNN) algorithms. The LR model demonstrated the best performance in predicting both SI and SA, with an accuracy of 0.74 (for both), specificity of 0.71 for SI; 0.69 for SA, and sensitivity (0.77 for SI; 0.81 for SA). It also achieved high positive predictive values (PPV) of 0.90 for SI; 0.95 for SA, and an area under the Receiver Operating Characteristic (AUROC) curve of 0.82 for both outcomes. The model identified cyberbullying (electronically bullied), drug use, forced sexual intercourse, sexual violence, difficulty in concentrating, and early use of alcohol or marijuana, as some of the most important and common predictors of SI and SA. In conclusion, our research advocates for the urgent development of support systems that address the early signs of suicidal behaviors. It also surfaces the fact that predicting with certainty who is likely to attempt suicide remains a challenge.

## Introduction

The question of 'why' often lingers after the loss of a loved one to suicide, accompanied by reflections on moments missed and support unoffered. This reality underscores the complex and often unseen nature of mental health struggles. It is often challenging to understand when

**Data Availability Statement:** The datasets used in the study can be retrieved from https://www.cdc.gov/healthyyouth/data/yrbs/data.htm.

**Funding:** The authors received no specific funding for this work.

**Competing interests:** The authors have declared that no competing interests exist.

suicidal thoughts set in, when people plan about it, and when they execute it—the deepest struggles often lie hidden beneath the surface, leaving friends and family wishing for a sign or a clue that they might have missed.

Suicide is now the 2nd leading cause of death, and numbers continue to rise [1]. According to the Center of Disease Control and Prevention (CDC), suicide rates increased 37% between 2000–2018 reaching its peak in 2021 [2]. According to the 2021 Youth Risk Behavior Surveillance System (YRBSS), approximately 10% of high school students attempted suicide and 22% seriously thought about it [3] in the United States. In 2022, about 49,449 people died by suicide in the United States [2]. The increasing prevalence of suicide among youth and its progression from ideation to attempt poses a significant challenge for public health globally [1, 4–9]. Despite substantial investments in research aimed at comprehending the risk factors for suicide and in developing suicide prevention initiatives [10], the incidence of suicide deaths among young Americans has risen in recent years [11, 12]. Studies have acknowledged poor mental health [13], encounters with violence [14], and social pressures [15, 16], as plausible factors influencing suicidal ideation and attempt among youth.

Machine learning (artificial intelligence) offers a promising approach to identifying individuals at risk of suicidal ideation or attempts, which is crucial for preventing suicide [17]. However, a significant limitation of many current models is their dependence on clinical data or psychological factors that are not easily accessible [18–20]. These models require specific data points, like the Patient Health Questionnaire (PHQ) depression score, that are typically only available through medical or psychological evaluations. The reliance on clinical data limits the models' practical use for the public, who may not have access to such detailed assessments. For example, if someone is worried about a loved one's mental health, they likely won't have the necessary clinical mental health assessment scores to use these predictive models effectively. This issue highlights a critical gap in the utility of these models for everyday situations and underscores the challenge of understanding at-risk behaviors in a domestic setting. The limitation points to a critical need for developing models that can operate with more commonly observable data points.

Our study aims to identify and leverage social and behavioral predictors of suicidal ideation and attempts among adolescents and young adults. Note that the objective is not to introduce a new predictive model as a replacement of clinical assessment but to assess if social and behavioral predictors that are observable by friends, family, or individuals themselves, can help predict suicidal ideation or attempt that a layperson can use as a preliminary tool at home.

## Method

### Ethics statement

The research is a secondary analysis of publicly available data and does not require approval from the institutional review board. The dataset contains no identifiers.

### Data description

We used the 2021 YRBSS as the data source [21]. YRBSS is a survey developed by the CDC that tracks behaviors leading to poor health in students grades 9 through 12. The surveys are administered every other year. Some of the health-related behaviors and experiences captured through the survey are student demographics, health behaviors, substance use behaviors, and student experiences. In 2021, YRBSS aimed to gather data from schools and students across the United States. Participation rates varied, with 152 out of the 209 schools agreeing to participate, yielding a school response rate of approximately 72.70%. In total 17,508 students completed the survey. After reviewing the responses for accuracy and completeness, 17,232

responses were deemed usable, leading to a student response rate of around 79.10%. To calculate the overall response rate for the survey, the response rates from both schools and students were combined, resulting in a final participation rate of about 57.50%.

Two variables, each indicative of suicidal ideation (SI) and suicidal attempts (SA), respectively, from the YRBSS were used as an outcome variable (target class) in our study.

*Suicidal Ideation (SI)*–The variable was measured using the following question: During the past 12 months, did you ever seriously consider attempting suicide? The responses were recorded as 'Yes' or 'No'. In our model, Yes was coded as 1.

*Suicidal Attempt (SA)*–The variable was measured using the following question: During the past 12 months, how many times did you actually attempt suicide? The responses were recorded as 0 times, 1 time, 2 or 3 times, 4 or 5 times, 6 or more times. In our study, we recorded all responses indicating at least one attempt as 1 and remaining as 0.

## Data preprocessing

Analyses were done in R (version 4.1.0). First, we duplicated and divided the data set into two data files, each having SI and SA as target variable, respectively. In this manuscript, we will address these datasets (D) as D-SA and D-SI. We then conducted correlation analysis on both the datasets (D-SI and D-SA) and removed all variables that were highly correlated either with the target class or among each other. As a result, we eliminated variables pertaining to sadness or hopelessness, suicide planning, and other measures of mental health. Missing values in target class were also deleted from the dataset. We then handled missing values across all input variables by mode replacement. The impute_data function [22] was used from dplyr package in R [23]. Categorical variables were one-hot encoded using dummyVars function [24]. Since the data consisted of variables with varying magnitude, we normalized the dataset [25].

Each of the preprocessed datasets (D-SI and D-SA) were divided into training and testing sets using a sample.split function [26] with an 80–20 ratio. To resolve the class imbalance problem, we used the oversampling of the minority class in the training set (D-SA and D-SI, respectively) by using Ovun.sample function [27]. Class imbalance, where there are many more negative instances (no ideation or attempt) than positive ones (presence of ideation or attempt), is common in sensitive issues like suicidal ideation (D-SI) and attempts (D-SA).

## Model selection

We compared three algorithms, extreme gradient boosting (XGB) [28], logistic regression (LR) [29], and K-Nearest neighbors (KNN) [30] on both D-SI and D-DS. We chose these models because of their performance in binary classification and computational efficiency [31–34]. XGB was selected for its performance and feature importance capabilities, offering insightful predictions even with the sparse data characteristic of survey responses. It excels in managing missing values and identifying key variables influencing suicidal ideation and attempts, when compared to models like random forest, which is more prone to overfitting in such a data-rich environment. Logistic regression was selected for its simplicity and interpretability. Lastly, KNN's flexibility and non-parametric nature allow it to adapt to the patterns within the YRBSS dataset, a feature not as directly accessible in more complex models that might struggle with the interpretability and computational demands presented by the dataset.

## Model training and evaluation

For the XGB model from XGBoost package [35], we converted the data (oversampled training data) into a matrix format, excluding the target class (SI for D-SI and SA for D-SA) variable. We used hyperparameter tuning to optimize the model. The best parameters included 0.1

learning rate (eta), 2 maximum depth of trees (max_depth), 1 n_thread and 0.1 gamma for regularization. For the LR model, we used glm function [36] with a binomial setting. Following the model fitting, predicted probabilities were generated, which were then converted to binary classifications using a 0.5 threshold. KNN model was used from Caret package [37].

All the models were tuned using 10-fold cross-validation. To address data dimensionality issues, we used xgb.importance function from the XGBoost package [35] to identify the most important variables for both D-SA and D-SI. All the models were retrained using only the important variables. Model evaluation was achieved by monitoring accuracy, sensitivity, specificity, positive predictive value (PPV), negative predictive value (NPV), and the area under the receiver operating characteristic curve (AUROC) [38].

## Odds ratio

Once the important variables were identified from our ML model, we used JASP [39] for calculating the prevalence odds ratio and 95% confidence interval of the top 10 predictors of ideation and attempts. Likert scale questions were converted to binary format for the analysis. For each question, if a response indicated the occurrence of an event or an action being taken at least once, it was assigned a value of 1; if no event or action was indicated, it was assigned a value of 0.

## Results

### Descriptive statistics

YRBSS consisted of 17,232 participants, of which 48% were female and 51.9% were male. All participants were students from grades 9 to 12 in the United States. Among respondents, 20.85% (3,593) reported suicidal ideation, and 10.17% (1,753) attempted suicide. Of the 3,526 participants with depression during covid, 46.23% (1,630) experienced suicidal ideation, and 19.46% (686) attempted suicide. Among the 8,152 female participants, 27.42% (2,335) reported suicidal ideation, and 13.37% (1,090) reported suicide attempts. Also, 29.06% of 3,981 participants who were overweight experienced suicide ideation and 14.02% of them attempted suicide. The other important information about the participants in the study is given in Table 1.

### Feature importance

Table 2 lists the top 10 important predictors of SI and SA, respectively. For SI, being 'electronically bullied' tops the list, underscoring the significant impact of cyberbullying on mental health. Following closely are factors such as the 'use of electronic vapor products' and 'serious difficulty in concentrating, remembering, or making decisions.' These might initially seem less directly connected to suicide ideation compared to more overtly distressing experiences like 'physical sexual violence', yet their prominence highlights the complex nature of mental health risks. Variables like 'sex', experience of 'being bullied on school property', and early initiation of 'alcohol use' were also significant, suggesting that both behavioral patterns and experiences of violence are critical in understanding suicide ideation.

For suicide attempts, the sequence starts similarly with 'electronic bullying' but is immediately followed by more severe experiences such as 'being physically forced to have sexual intercourse'. This indicates a stronger link between direct, physical forms of violence and the act of attempting suicide compared to ideation alone. The 'use of electronic vapor products' remains highly relevant, suggesting a possible link between substance use and the escalation from ideation to attempts. The list for attempts also includes more specific variables like 'early use of alcohol and drugs', 'current marijuana use', and experiences of 'physical dating violence',

**Table 1. Participant characteristics and socio-demographics.**

| | Sample size (n = 17232) | Suicide ideation | Suicide attempt |
|---|---|---|---|
| | | 3593 (20.86%) | 1753 (10.18%) |
| **Age** | | | |
| Under 18 years old | 16,111 | 3377 (20.97%) | 1641 (10.19%) |
| 18 years old or older | 1,023 | 196 (19.16%) | 100 (9.78%) |
| **Biological sex** | | | |
| Male | 8,816 | 1151 (13.06%) | 603 (6.84%) |
| Female | 8,152 | 2335 (27.42%) | 1090 (13.37%) |
| **School grade level** | | | |
| 9[th] grade | 4,646 | 937 (20.17%) | 522 (11.24%) |
| 10[th] grade | 4,466 | 988 (22.12%) | 501 (11.22%) |
| 11[th] grade | 4,118 | 887 (21.54%) | 376 (9.13%) |
| 12[th] grade | 3,843 | 755 (19.65%) | 326 (8.48%) |
| Ungraded or another grade | 23 | 7 (30.43%) | 9 (39.13%) |
| **Race** | | | |
| Hispanic or Latino | 3,258 | 670 (20.56%) | 352 (10.80%) |
| American Indian or Alaska Native | 801 | 226 (28.21%) | 116 (14.48%) |
| Asian | 1,330 | 278 (20.90%) | 134 (10.08%) |
| Black or African American | 3,120 | 688 (22.05%) | 435 (13.94%) |
| Native Hawaiian or Other Pacific Islander | 391 | 99 (25.32%) | 61 (15.60%) |
| White | 11,269 | 2455 (21.79%) | 1077 (9.56%) |
| **Feeling stress, anxiety, and depression during Covid-19** | | | |
| Never or rarely | 3,795 | 253 (6.67%) | 121 (3.19%) |
| Sometimes | 2,395 | 335 (13.99%) | 115 (4.80%) |
| Most of the times or always | 3,526 | 1630 (46.23%) | 686 (19.46%) |
| **Feeling sad or hopeless, during the past 12 months** | | | |
| Yes | 6,749 | 3,114 (46.14%) | 1,382 (20.48%) |
| No | 10,212 | 462 (4.52%) | 328 (3.21%) |
| **Initiation of cigarette smoking** | | | |
| Have never smoked | 12,947 | 2,221(17.15%) | 944 (7.29%) |
| Before 13 years old | 1,074 | 499 (46.46%) | 344 (32.03%) |
| After 13 years old | 1,661 | 572 (34.44%) | 297 (17.88%) |
| **currently smoked cigarette** | | | |
| Yes | 659 | 332 (50.38%) | 253 (38.39%) |
| No | 16,253 | 3,181 (19.57%) | 1,441 (8.87%) |
| **Initiation of alcohol use** | | | |
| Have never used | 8,742 | 1,027 (11.75%) | 434 (4.96%) |
| Before 13 years old | 2,499 | 987 (39.50%) | 582 (23.29%) |
| After 13 years old | 5,272 | 1,450 (27.50%) | 650 (12.33%) |
| **Currently drank alcohol** | | | |
| Yes | 3648 | 1,308 (35.86%) | 725 (19.87%) |
| No | 12,586 | 2,028 (16.11%) | 867 (6.89%) |
| **Initiation of marijuana use** | | | |
| Have never used | 12,275 | 1,828 (14.89%) | 770 (6.27%) |
| Before 13 years old | 873 | 406 (46.51%) | 301 (34.48%) |
| After 13 years old | 3,662 | 1,266 (34.57%) | 617 (16.85%) |
| **Currently used marijuana** | | | |
| Yes | 2,647 | 1,109 (41.90%) | 638 (24.10%) |

*(Continued)*

**Table 1.** (Continued)

| | Sample size (n = 17232) | Suicide ideation | Suicide attempt |
|---|---|---|---|
| | | 3593 (20.86%) | 1753 (10.18%) |
| No | 14,250 | 2,409 (16.91%) | 1,061 (7.45%) |
| **Ever used an electronic vapor product** | | | |
| Yes | 6,045 | 2,050 (33.91%) | 1,073 (17.75%) |
| No | 10,761 | 1,462 (13.59%) | 599 (5.57%) |
| **Currently used an electronic vapor product** | | | |
| Yes | 2,922 | 1,199 (41.03%) | 700 (23.96%) |
| No | 13,155 | 2,071 (15.74%) | 858 (6.52%) |
| **Initiation of sexual intercourse** | | | |
| Have never had | 10,742 | 1,760 (16.38%) | 708 (6.59%) |
| Before 13 years old | 528 | 271 (51.33%) | 196 (37.12%) |
| After 13 years old | 4,229 | 1,234 (29.18%) | 621 (14.68%) |
| **Currently sexually active** | | | |
| Yes | 3,347 | 1,039 (31.04%) | 583 (17.42%) |
| No | 12,136 | 2,221 (18.30%) | 940 (7.75%) |
| **Experienced sexual violence (kissing, touching, or being physically forced to have sexual intercourse)** | | | |
| Yes | 1,479 | 829 (56.05%) | 482 (32.59%) |
| No | 11,858 | 1,880 (15.85%) | 653 (5.51%) |
| **Experienced physical dating violence, during the past 12 months** | | | |
| Yes | 837 | 467 (55.79%) | 347 (41.46%) |
| No | 15,807 | 3,041 (19.24%) | 1,337 (8.46%) |
| **Forced sexual intercourse** | | | |
| Yes | 1,193 | 739 (61.94%) | 460 (38.56%) |
| No | 12,965 | 2,207 (17.02%) | 783 (6.04%) |
| **Ever had sexual contact with both male and female** | | | |
| Yes | 783 | 485 (61.94%) | 281 (35.89%) |
| No | 12,836 | 2,393 (18.64%) | 930 (7.25%) |
| **Electronically bullied, during the past 12 months** | | | |
| Yes | 2,765 | 1,251 (45.24%) | 703 (25.42%) |
| No | 14,265 | 2,326 (16.31%) | 1,018 (7.14%) |
| **Bullying at school, during the past 12 months** | | | |
| Yes | 2,712 | 1,181 (43.55%) | 534 (19.69%) |
| No | 13,994 | 2,323 (16.60%) | 1,052 (7.52%) |
| **Safety concerns at school, during the past 30 days** | | | |
| Yes | 1,394 | 600 (43.04%) | 403 (28.91%) |
| No | 15,716 | 2,962 (18.85%) | 1,320 (8.40%) |
| **Threatened or injured with a weapon at school, during the past 12 months** | | | |
| Yes | 1,118 | 497 (44.45%) | 353 (31.57%) |
| No | 15,560 | 2,980 (19.15%) | 1,344 (8.64%) |
| **Were offered, sold, or given an illegal drug on school property, during the past 12 months** | | | |
| Yes | 2,265 | 877 (38.72%) | 479 (21.15%) |
| No | 14,101 | 2,555 (18.12%) | 1,174 (8.33%) |
| **Ever prescription pain medicine use, without a doctor's prescription or differently than how a doctor told them to use it** | | | |
| Yes | 2,047 | 937 (45.77%) | 557 (27.21%) |
| No | 14,810 | 2,582 (17.43%) | 1,146 (7.74%) |
| **Carried a gun, during the past 12 months** | | | |
| Yes | 588 | 205 (34.86%) | 183 (31.12%) |

(*Continued*)

**Table 1.** (Continued)

| | Sample size (n = 17232) | Suicide ideation | Suicide attempt |
|---|---|---|---|
| | | 3593 (20.86%) | 1753 (10.18%) |
| No | 12,488 | 2,670 (21.38%) | 1,264 (10.12%) |
| **Physical activity at least 1 hour in a day, during the past 7 days** | | | |
| Have not had | 2,626 | 715 (27.23%) | 321 (12.22%) |
| 4 or less days | 6,368 | 1,499 (23.54%) | 746 (11.71%) |
| 5 or more days | 7,658 | 1,271 (16.60%) | 614 (8.02%) |
| **Hours of sleep on school nights** | | | |
| 4 hours or less | 1,391 | 564 (40.55%) | 306 (22.00%) |
| 5 to 7 hours | 9,005 | 1828 (20.30%) | 683 (7.58%) |
| 8 hours or more | 3,219 | 433 (13.45%) | 207 (6.43%) |
| **Difficulty concentrating, remembering, or making decisions** | | | |
| Yes | 4,130 | 1636 (39.61%) | 699 (16.92%) |
| No | 4,921 | 427 (8.68%) | 149 (3.03%) |

painting a picture of a cumulative risk landscape where substance use, and violent experiences significantly contribute to the likelihood of an attempt. Overall, the results emphasize the point that while some factors are critical in contemplating suicide, others play a more crucial role in the progression to actual attempts, underlining the necessity of a nuanced approach to prevention that addresses both immediate and underlying risk factors.

## Model outcomes

Table 3 reports the model performance. For SI, LR shows the highest accuracy (0.74) and specificity (0.71), and it has competitive values for sensitivity (0.77), PPV (0.91), and AUROC (0.82). Its NPV is the lowest among the models for SI, but this is by a marginal difference. The XGB model has slightly lower accuracy and specificity but higher sensitivity, indicating it's better at identifying true positives but slightly worse overall. The KNN model lags in all aspects except for sensitivity, where it is the lowest, indicating it's generally less effective for this prediction task.

**Table 2. Top 10 most important predictors of suicidal ideation and suicidal attempts.**

| Suicidal ideation | Suicidal attempt |
|---|---|
| Electronically bullied | Electronically bullied |
| Used electronic vapor product | Physically forced to have sexual intercourse |
| Serious difficulty concentrating, remembering, or making decisions | Used electronic vapor product |
| Experienced sexual violence | Took prescription pain medicine without a doctor's prescription or differently than how a doctor told them to use it |
| Took prescription pain medicine without a doctor's prescription or differently than how a doctor told them to use it. | Experienced sexual violence |
| Biological sex | Use an electronic vapor product (current use) |
| Bullied on school property | First drink of alcohol before age 13 years. |
| Use marijuana | Marijuana for the first time before age 13 years |
| Physically forced to have sexual intercourse | Use marijuana |
| First drink of alcohol before age 13 years. | Physical dating violence |

**Table 3. Performance measures for ML models.**

| Predictors | Model | Accuracy | Sensitivity | Specificity | PPV | NPV | AUROC |
|---|---|---|---|---|---|---|---|
| Suicide ideation | XGB | 0.73 | 0.78 | 0.68 | 0.90 | 0.46 | 0.81 |
| | LR | 0.74 | 0.77 | 0.71 | 0.91 | 0.46 | 0.82 |
| | KNN | 0.69 | 0.73 | 0.65 | 0.89 | 0.39 | 0.73 |
| Suicide attempt | XGB | 0.72 | 0.81 | 0.62 | 0.94 | 0.31 | 0.81 |
| | LR | 0.74 | 0.81 | 0.68 | 0.95 | 0.32 | 0.82 |
| | KNN | 0.64 | 0.83 | 0.44 | 0.92 | 0.26 | 0.65 |

XGB: extreme gradient boosting; LR: logistic regression; KNN: K-nearest neighbors; PPV: positive predictive value; NPV: negative predictive value; AUROC: area under the receiver operating characteristic curve.

For SA, the LR model again shows the highest accuracy (0.74) and specificity (0.68), and it has very competitive sensitivity (0.81) and the highest PPV (0.95) and AUROC (0.82). Despite a lower NPV, it demonstrates the best balance across all metrics. The XGB model, while having a slightly higher sensitivity, falls short in accuracy, specificity, and NPV, indicating it might not balance false positives and negatives as well as the LR model. The KNN model, despite its highest sensitivity for suicide attempts, has significantly lower accuracy, specificity, and the lowest NPV, suggesting it might misclassify a greater number of non-attempt cases as attempts. Considering these metrics, LR model emerges as the best fit model for both SI and SA predictions.

## Prevalence odds ratio

Table 4 shows the factors associated with increased odds of SI and SA. Electronic bullying emerged as a critical factor, with individuals who experienced electronic bullying showing higher odds of SI (aOR = 1.45) and SA (aOR = 2.52). Similarly, the use of electronic vapor products was associated with a modest increase in SI (aOR = 1.20) and a slightly elevated, though less certain, risk for SA (aOR = 1.14). Cognitive difficulties—specifically issues with concentration, memory, or decision-making—showed a notably strong association with SI (aOR = 4.23), indicating a significant mental health burden linked to SI risk. Individuals exposed to sexual violence demonstrated higher odds for both SI (aOR = 2.10) and SA (aOR = 2.21). Additionally, misuse of prescription pain medication was associated with increased odds of both SI (aOR = 1.95) and SA (aOR = 2.11). Gender differences were observed, with females showing a higher likelihood of SI than males (aOR = 1.50). Social factors, such as being bullied on school property, were also significantly associated with SI (aOR = 1.80). Early substance use, including marijuana and alcohol consumption before age 13, was linked to increased odds of both SI and SA, with early alcohol use (aOR = 1.63; 1.76) and marijuana use (aOR = 1.60; 1.52). Furthermore, experiencing physical dating violence was strongly associated with SA (aOR = 2.00), highlighting the impact of interpersonal violence on suicide-related behaviors.

## Discussion

### Overall contribution

While we acknowledge that the relationships between the predictors of SI and SA are well established in the literature, our study brings a fresh perspective to predicting thoughts of suicide and suicide attempts by focusing on observable, everyday behaviors and experiences of young people, such as experiences of bullying, drug use (including alcohol, cigarettes,

**Table 4. Prevalence odds ratio and %95 confidence interval of the top 10 most important predictors of suicidal ideation and suicidal attempts.**

| Suicide Ideation | OR [CI 95%] | Suicide Attempt | OR [CI 95%] |
|---|---|---|---|
| Electronically bullied | 1.45 [0.20, 0.55] | Electronically bullied | 2.52 [0.76, 0.90] |
| Used electronic vapor product | 1.20 [0.03, 0.33] | Physically forced to have sexual intercourse | 2.87 [0.85, 1.26] |
| Serious difficulty concentrating, remembering, or making decisions | 4.23 [1.31, 1.58] | Used electronic vapor product | 1.14 [-0.12, 0.37] |
| Experienced sexual violence | 2.10 [0.56, 0.93] | Took prescription pain medicine without a doctor's prescription or differently than how a doctor told them to use it | 2.11 [0.56, 0.93] |
| Took prescription pain medicine without a doctor's prescription or differently than how a doctor told them to use it. | 1.95 [0.50, 0.84] | Experienced sexual violence | 2.21 [0.60, 0.99] |
| Biological sex | 1.5 [0.28, 0.53] | Use an electronic vapor product (current use) | 1.25 [-0.02, 0.46] |
| Bullied on school property | 1.80 [0.42, 0.76] | First drink of alcohol before age 13 years. | 1.76 [0.36, 0.77] |
| Use marijuana | 1.60 [0.31, 0.63] | Marijuana for the first time before age 13 years | 1.29 [-0.00, 0.51] |
| Physically forced to have sexual intercourse | 2.15 [0.55, 0.98] | Use marijuana | 1.52 [0.18, 0.67] |
| First drink of alcohol before age 13 years. | 1.63 [0.34, 0.64] | Physical dating violence | 2.00 [0.43, 0.96] |

marijuana, and vapor products), experiences of sexual and physical violence, and difficulties with concentration (like remembering things or making decisions). Unlike many existing studies (Table 5), which rely heavily on clinical or latent constructs such as severe depression, anxiety, previous suicide attempts, antisocial score, or perceived warmth, our approach uses data that are more accessible and observable by public, like teachers, parents, or friends. This makes our model more practical for real-world prevention efforts. Besides, the essence of using predictive models in suicide prevention is to uncover hidden patterns that aren't immediately obvious to humans. If we only used variables that clearly indicate a high risk of reattempt, such as a history of suicide attempts or severe depression, we wouldn't be fully utilizing preventive approaches. Such information, while crucial, might not help in early prevention because it's already an established risk factor. Our study reveals new insights that can guide prevention strategies. For example, our model, using the YRBSS, identified predictors like being electronically bullied, using an electronic vapor product, experiencing sexual violence, and misusing prescription pain medicine, achieving an AUROC of 0.82 for suicidal ideation and 0.83 for suicide attempts, with an accuracy of 0.74 for both. In contrast, other studies, such as those using the Health Research Enterprise from French universities [40] or the Minnesota Multiphasic Personality Inventory-2 Restructured Form [41], focus on predictors like suicidal thoughts, anxiety, depression, and emotional dysfunction, achieving similar or slightly higher AUROC scores but often relying on less observable factors. Therefore, our model's emphasis on accessible data offers a significant advantage in practical applications for early intervention and prevention efforts.

## Electronic bullying as a risk factor

We identified electronic bullying (cyberbullying) as a key factor, with individuals who experienced it showing higher odds of both suicidal ideation and attempts. This form of harassment

**Table 5. Comparison of our model's performance with other recent ML studies for predicting risk of suicide.**

| | Dataset (name and size) | Important predictors | Target class | Final models | Final model's performance |
|---|---|---|---|---|---|
| Our model | Youth Risk Behavior Surveillance System | electronically bullied; used an electronic vapor product; experienced sexual violence; and ever took prescription pain medicine without a doctor's prescription or differently than how a doctor told them to use it | SA; SI | LR | *SI; SA* |
| | | | | | AUROC: 0.82; 0.83 |
| | | | | | Accuracy: 0.74; 0.74 |
| | *N = 17,232* | | | | Sensitivity: 0.77; 0.81 |
| [40] | Health Research Enterprise from French universities | **suicidal thoughts;** self-esteem; trait **anxiety; depression;** perceived **stress** | SA; SI | RF | |
| | | | | | AUROC: 0.80 |
| | *N = 5,066* | | | | Sensitivity: 0.79 |
| [41] | Minnesota Multiphasic Personality Inventory-2 Restructured Form | **emotional dysfunction; stress; anxiety;** family problems; aggressiveness; shyness. | SA; SI | RF | *SI; SA* |
| | | | | | AUROC: 0.84; 0.85 |
| | *N = 7,824* | | | | Accuracy: 0.92; 0.95 |
| [31] | Korea Youth Risk Behavior Web-based Survey | **sadness;** violence; substance use; **stress.** | SA; SI | XGB | |
| | | | | | AUROC: 0.86 |
| | *N = 59,984* | | | | Accuracy: 0.79 |
| [19] | Korea National Health & Nutrition Examination Survey | depressed mood over two weeks; **stress** level in daily life; **anxiety/depression;** sex; education. | SI | RF | |
| | | | | | AUROC: 0.85 |
| | | | | | Accuracy: 0.821 |
| | *N = 11,628* | | | | Sensitivity: 0.836 |
| [42] | Medical students from three medical schools across China | **suicidal ideation; suicide plan;** anxiety; depression; and relationship with the participant's father. | SA | RF | |
| | | | | | AUROC: 0.925 |
| | | | | | Accuracy: 0.90 |
| | *N = 4,882* | | | | Sensitivity: 0.73 |
| [43] | Early Developmental Stages of Psychopathology | **prior suicide attempt;** education; prior help seeking; parental loss or separation. | SA | N/A | |
| | | | | | AUROC: 0.82–0.82 |
| | | | | | Sensitivity: 0.02–0.25 |
| | *N = 2,797* | | | | |
| [44] | Québec Longitudinal Study of Child Development | adulthood mother **antisocial score;** adolescent father antisocial score; mother **perceived warmth and affection;** adulthood father antisocial score, family size | SA | RF | |
| | | | | | AUROC: 0.72 |
| | | | | | Sensitivity: 0.50 |
| | *N = 1,623* | | | | |
| [45] | Ungdata surveys | felt **depressed;** felt worthless; disappointed with myself; contact with a **psychologist/psychiatrist.** | SA | XGB | AUROC: 0.92 |
| | *N = 173,664* | | | | Sensitivity: 0.77 |

*(Continued)*

**Table 5.** (Continued)

| | Dataset (name and size) | Important predictors | Target class | Final models | Final model's performance |
|---|---|---|---|---|---|
| [46] | UK Biobank | **psychiatric disorders; anxiety; depression; hopelessness;** age; history of **suicide attempts**. | SA | Light Gradient Boosting | *Within 1 year*: |
| | | | | | AUROC: 0.901 |
| | | | | | Sensitivity: 0.605 |
| | *N = 50,310* | | | | *Between 1 to 6 years*: |
| | | | | | AUC: 0.885 |
| | | | | | Sensitivity: 0.558 |
| [47] | Korea Welfare Panel Study | **mental health;** economic condition; satisfaction with family relations; life satisfaction; smoking; mother's educational background. | SA; SI | XGB | *SI; SA* |
| | | | | | AUROC: 0.86–0.88 |
| | *N = 60,568* | | | | Accuracy: 0.86–0.88 |
| | | | | | Sensitivity: 0.85–0.85 |
| [48] | Korea Youth Risk Behavior Survey | **suicide ideation; suicide plan;** sex; grade; academic achievement; family structure; education of father; education of mother; current smoking; current alcohol drinking; drug experience; **sadness, hopelessness** | SA | LR; ANN; XGB | AUROC: 0.94–0.95 |
| | *N = 468,482* | | | | Accuracy: 0.97 |
| | | | | | |
| [49] | Ten psychological scale scores and 20 sociodemographic parameters of Chinese adolescents | **depression;** history of **suicidal thoughts;** lack of motivation; lack of social support. | SI | RF | |
| | | | | | AUROC: 0.92 |
| | *N = 10,243* | | | | Accuracy: 0.87 |

often extends beyond the school environment into the digital lives of young individuals [50], it poses a constant threat to their well-being [51]. Research has shown that victims of cyberbullying, when compared to non-victims, have a higher risk of suicidal ideation, self-harm, and suicide attempts [52]. Additionally, exposure to both cyberbullying and traditional in-school bullying has been associated with depressive symptoms, gradually increasing the risk for suicidal ideation, suicide planning, and suicide attempts [53]. Overall, our research underscores the detrimental impact of bullying, especially cyberbullying, on adolescent mental health, highlighting the need for targeted interventions to address bullying behaviors and support at-risk individuals to prevent suicidal ideation and attempts.

## Drug use as a risk factor

Another predictor of suicide ideation and attempts identified in our model is drug use. We noted adolescent indulged with misuse of prescription medicine, marijuana and alcohol use during early and pre-teen age had higher odds of ideation and attempt. Drug use has been consistently linked to an increased risk of suicidal ideation and suicide attempts among various populations [54]. Studies have shown a strong association between illicit drug use and suicidal ideation, self-harm, and suicide attempts [55]. Early initiation of substance use, including smoking cigarettes, alcohol, and drugs like cannabis, has been correlated with suicidal ideation and attempts among adolescents [56]. Additionally, while some research suggests that drug use, particularly marijuana and other drugs, may not directly increase the risk of suicidal ideation, there is a bidirectional relationship where suicidal ideation can lead to an increased risk of illicit drug use [57].

### Sexual misconduct as a risk factor

We identified forced sexual encounters and sexual violence as key factors, with adolescents who were physically forced to have sexual intercourse, experienced sexual violence, or physical dating violence, had increased odds of ideation and or attempt. In line with our finding, a recent study in 2024, reported strong association between sexual harassment and suicidal ideation [58]. Additionally, the concurrence of sexual violence has been independently associated with an elevated risk of suicidal behaviors among adolescents [59]. Contrary to some expectations, our findings and 2014 study noted bullying to have a stronger effect on young people's suicidal ideation compared to sexual abuse or physical violence [60]. Another variable our model acknowledged as a predictor of suicidal ideation is difficulty in concentrating, remembering, or making decisions. The impact of cognitive deficits on decision-making processes has been identified as a significant factor influencing suicidal ideation [61]. Additionally, the inability to suppress unwanted thoughts or lack of executive functions has been recognized as contributing factors to suicidal ideation.

### Implications

Our analysis sheds light on early signs of suicide ideation and attempts among youth, offering a plausible pathway for suicide prevention. Identifying electronic bullying as a prominent risk factor underscores its pervasive threat to adolescent mental health. This insight suggests that effective suicide prevention strategies must address the digital landscape where young individuals spend a significant portion of their lives. Interventions aimed at combating cyberbullying could serve as a critical early step in safeguarding against the progression towards suicidal thoughts and behaviors. Furthermore, the association between drug use and increased risk of suicide ideation and attempts highlights the importance of substance abuse prevention as a cornerstone of suicide prevention efforts. Early intervention programs that educate young people about the dangers of substance use and provide support for those struggling could significantly reduce the risk of suicide. Additionally, recognizing signs of forced sexual encounters and sexual violence as predictors of suicide ideation points to the need for accessible support systems and trauma-informed care for victims. Addressing cognitive difficulties, such as problems with concentration, memory, or decision-making, further expands our understanding of suicide prevention, emphasizing the importance of comprehensive mental health support that includes cognitive and emotional aspects. By focusing on these early signs and implementing targeted prevention strategies, we can create a more effective approach to reducing the incidence of suicide ideation and attempts among youth.

### Limitations

Our study has few limitations that should be considered. First, the use of a single item to determine suicidal ideation may overestimate its prevalence, potentially affecting the accuracy and reliability of our predictions. Second, while the YRBSS provides a comprehensive dataset, it relies on self-reported data, which may introduce biases such as underreporting or misreporting of sensitive behaviors and experiences. Third, our model was developed and validated using data from students in grades 9 to 12 in the United States, which may limit the generalizability of our findings to other populations or age groups. Additionally, the cross-sectional nature of the YRBSS data prevents us from making causal inferences about the specific directional causal relationships between the predictors and suicidal behaviors. Finally, our study focuses on social and behavioral factors, which, while important, do not encompass all potential risk factors for suicidal ideation and attempts, such as genetic, biological, or environmental

influences. Future research should consider integrating these additional dimensions to provide a more holistic understanding of the predictors of suicidal behaviors.

## Conclusion

Our study provides critical insights into the predictors of suicidal ideation and attempts among youth, highlighting the significant impact of factors such as cyberbullying, drug use, sexual violence, and misuse of prescription medications. By understanding these influences, families are better equipped to recognize warning signs and intervene effectively, underscoring the need to create supportive home environments that promote mental well-being and resilience. Additionally, this research advocates for the urgent development of support systems that address the early signs of suicidal behaviors. Aligning with the widely accepted conclusion, our study also surfaces the fact that predicting with certainty who is likely to attempt suicide remains a challenge.

## Author Contributions

**Conceptualization:** Avishek Choudhury.

**Data curation:** Yeganeh Shahsavar, Avishek Choudhury.

**Formal analysis:** Yeganeh Shahsavar, Avishek Choudhury.

**Investigation:** Avishek Choudhury.

**Methodology:** Avishek Choudhury.

**Resources:** Avishek Choudhury.

**Software:** Avishek Choudhury.

**Supervision:** Avishek Choudhury.

**Validation:** Avishek Choudhury.

**Writing – original draft:** Yeganeh Shahsavar, Avishek Choudhury.

**Writing – review & editing:** Avishek Choudhury.

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
