## [Editor Report · Decision Letter 0]

26 Jul 2024

PMEN-D-24-00254

Behavioral and Social Predictors of Suicidal Ideation and Attempts Among Adolescents and Young Adults

PLOS Mental Health

Dear Dr. Choudhury,

Thank you for submitting your manuscript to PLOS Mental Health. After careful consideration, we feel that it has merit but does not fully meet PLOS Mental Health’s publication criteria as it currently stands. Therefore, we invite you to submit a revised version of the manuscript that addresses the points raised during the review process.

We look forward to receiving your revised manuscript.

Kind regards,

Justus Uchenna Onu, MBBS, FWACP, FMCPsych

Academic Editor

PLOS Mental Health

Journal Requirements:

Additional Editor Comments (if provided):

Editor’s comments

Thank you for submitting your manuscript to this journal. Kindly address the following comments:

Abstract:

Conclude based on your findings

Background:

Well-focused, provided context but requires grammatical editing

Methods:

• Specify what age range was considered in this study

• The use of only item to determine suicidal idea may overestimate it and may be considered a potential limitation

• The grammatical errors persisted in the method section, kindly carry out grammar editing

Results:

Well-presented

Discussion:

How comparable is your findings to the global literature

Limitation:

List potential limitations of this study

Conclusion:

Stick to your findings
---

## [Decision Letter · Decision Letter 1]

22 Oct 2024

PMEN-D-24-00254R1

Behavioral and Social Predictors of Suicidal Ideation and Attempts Among Adolescents and Young Adults

PLOS Mental Health

Dear Dr. Choudhury,

Thank you for submitting your manuscript to PLOS Mental Health. After careful consideration, we feel that it has merit but does not fully meet PLOS Mental Health’s publication criteria as it currently stands. Therefore, we invite you to submit a revised version of the manuscript that addresses the points raised during the review process.

We look forward to receiving your revised manuscript.

Kind regards,

Justus Uchenna Onu, MBBS, FWACP, FMCPsych

Academic Editor

PLOS Mental Health

Journal Requirements:

Additional Editor Comments (if provided):

Thank you for your patience. We have now secured an additional review of your manuscript and you are now expected to make some minor revision.

Reviewers' comments:

Reviewer's Responses to Questions

**Comments to the Author**

1. If the authors have adequately addressed your comments raised in a previous round of review and you feel that this manuscript is now acceptable for publication, you may indicate that here to bypass the “Comments to the Author” section, enter your conflict of interest statement in the “Confidential to Editor” section, and submit your "Accept" recommendation.

Reviewer #1: All comments have been addressed

Reviewer #2: All comments have been addressed

2. Does this manuscript meet PLOS Mental Health’s publication criteria? Is the manuscript technically sound, and do the data support the conclusions? The manuscript must describe methodologically and ethically rigorous research with conclusions that are appropriately drawn based on the data presented.

Reviewer #1: Yes

Reviewer #2: Yes

3. Has the statistical analysis been performed appropriately and rigorously?

Reviewer #1: Yes

Reviewer #2: Yes

4. Have the authors made all data underlying the findings in their manuscript fully available (please refer to the Data Availability Statement at the start of the manuscript PDF file)?

Reviewer #1: Yes

Reviewer #2: Yes

5. Is the manuscript presented in an intelligible fashion and written in standard English?

Reviewer #1: Yes

Reviewer #2: Yes

6. Review Comments to the Author

Reviewer #1: This a well written manuscript.

Some the references do not have neither page numbers nor doi., and some web page cited do not have URL. Please go through the reference and make corrections.

Reviewer #2: Abstract:

• Line 2 should read “The study aimed to determine the social and behavioral predictors……………”

• Write the abbreviations in full before subsequently abbreviating e.g., PPV, AUROC etc.

“The model identified cyberbullying, drug use, and sexual violence, difficulty in 42 concentrating, early initiation of alcohol or marijuana, as some of the most important predictors”

• Predictors of what?

• Kindly insert the test statistics e.g., the odd ratio and its 95% confidence interval

• Keywords: Add “social” and “behavioral predictors”

Background

• Rephrase the objective in the last paragraph as suggested in the abstract

Methods

• Is appropriate

Results

• Add the 95% confidence interval for the prevalence of suicidal ideation and attempt

7. PLOS authors have the option to publish the peer review history of their article (what does this mean?). If published, this will include your full peer review and any attached files.

**Do you want your identity to be public for this peer review?** For information about this choice, including consent withdrawal, please see our Privacy Policy.

Reviewer #1: No

Reviewer #2: No

---

## [Decision Letter · Decision Letter 2]

2 Dec 2024

PMEN-D-24-00254R2

Behavioral and Social Predictors of Suicidal Ideation and Attempts Among Adolescents and Young Adults

PLOS Mental Health

Dear Dr. Choudhury,

Thank you for submitting your manuscript to PLOS Mental Health. As explained previously, I took over the handling of your paper in order to ensure that an additional review was secured prior to acceptance of the paper. Thank you for your understanding and patience with this process. I am pleased to say that, having now secured the final review, we would be happy to offer one final minor revision, which I will then assess in house for efficiency. You will be able to see the final reviewer comments at the end of this email. Please feel free to reach out to me directly should you have any questions at all.

We look forward to receiving your revised manuscript and thank you for your patience with this process and understanding. 

Kind regards,

Karli Montague-Cardoso

Executive Editor

PLOS Mental Health

Journal Requirements:

Additional Editor Comments (if provided):

Reviewers' comments:

Reviewer's Responses to Questions

**Comments to the Author**

1. If the authors have adequately addressed your comments raised in a previous round of review and you feel that this manuscript is now acceptable for publication, you may indicate that here to bypass the “Comments to the Author” section, enter your conflict of interest statement in the “Confidential to Editor” section, and submit your "Accept" recommendation.

Reviewer #3: All comments have been addressed

2. Does this manuscript meet PLOS Mental Health’s publication criteria? Is the manuscript technically sound, and do the data support the conclusions? The manuscript must describe methodologically and ethically rigorous research with conclusions that are appropriately drawn based on the data presented.

Reviewer #3: Yes

3. Has the statistical analysis been performed appropriately and rigorously?

Reviewer #3: I don't know

4. Have the authors made all data underlying the findings in their manuscript fully available (please refer to the Data Availability Statement at the start of the manuscript PDF file)?

Reviewer #3: Yes

5. Is the manuscript presented in an intelligible fashion and written in standard English?

Reviewer #3: Yes

6. Review Comments to the Author

Reviewer #3: The authors identify the behavioural and social predictors of suicidal ideation and atempts among adolescents and young adults in the US. This is an important piece that warrants publication in the journal. I have only two comments:

1. While poetic, the opening sentences in the study can be quite triggering for readers, especially for those with lived experience of suicide beravement. I would suggest changing it.

2. While the study uses a novel method of analysis, the relationships between the predictors and SI and SA are well established in the literature. Thus, I would urge the authors to temper their language of the novelty of the findings. This would serve well, given that until today, even machine learning methods struggle to predict who is likely to attempt suicide - leading to the widely accepted conclusion that utilising factors relating to suicide to prevent suicide are challenging.

7. PLOS authors have the option to publish the peer review history of their article (what does this mean?). If published, this will include your full peer review and any attached files.

**Do you want your identity to be public for this peer review?** For information about this choice, including consent withdrawal, please see our Privacy Policy.

Reviewer #3: No

---

## [Editor Report · Decision Letter 3]

11 Dec 2024

Behavioral and Social Predictors of Suicidal Ideation and Attempts Among Adolescents and Young Adults

PMEN-D-24-00254R3

Dear Dr. Choudhury,

Thank you for submitting your revision to us and for your understanding and patience with the process as we obtained another review. We apologise for the back-and-forth and appreciate that you continued with our journal. We are pleased to inform you that your manuscript 'Behavioral and Social Predictors of Suicidal Ideation and Attempts Among Adolescents and Young Adults' has been provisionally accepted for publication in PLOS Mental Health.

Best regards,

Karli Montague-Cardoso

Executive Editor

PLOS Mental Health